# High-Efficiency p-Type Si Solar Cell Fabricated by Using Firing-Through Aluminum Paste on the Cell Back Side

**DOI:** 10.3390/ma12203388

**Published:** 2019-10-17

**Authors:** Guang Wu, Yuan Liu, Mengxue Liu, Yi Zhang, Peng Zhu, Min Wang, Genhua Zheng, Guangwei Wang, Deliang Wang

**Affiliations:** 1College of Chemistry and Chemical Engineering, Nantong University, 226019 Nantong, China; SA226148@mail.ustc.edu.cn; 2Nano Science and Technology Institute, University of Science and Technology of China, 230026 Hefei, China; lmxlmx@mail.ustc.edu.cn (M.L.); zyjw@mail.ustc.edu.cn (Y.Z.); 3Hefei National Lab for Physical Sciences at Microscale, University of Science and Technology of China, 230026 Hefei, China; min@mail.ustc.edu.cn (M.W.); zghno1@mail.ustc.edu.cn (G.Z.); wangggw@mail.ustc.edu.cn (G.W.)

**Keywords:** firing-through paste, PERC, contact formation, open-circuit voltage

## Abstract

Firing-through paste used for rear-side metallization of p-type monocrystalline silicon passivated emitter and rear contact (PERC) solar cells was developed. The rear-side passivation Al_2_O_3_ layer and the SiN_x_ layer can be effectively etched by the firing-through paste. Ohmic contact with a contact resistivity between 1 to 10 mΩ·cm^2^ was successfully fabricated. Aggressive reactive firing-through paste would introduce non-uniform etching and high-density recombination centers at the Si/paste interface. Good balance between low resistive contact formation and relatively high open-circuit voltage can be achieved by adjusting glass frit and metal powder content in the paste. Patterned dot back contacts formed by firing-through paste can further decrease recombination density at the Si/paste interface. A P-type solar cell with an area of 7.8 × 7.8 cm^2^ with a V_oc_ of 653.4 mV and an efficiency of 19.61% was fabricated.

## 1. Introduction

In the last two decades the efficiency of large-area crystalline Si solar cells has been increased significantly. The efficiency increase was mainly ascribed to the new concepts introduced to the crystalline Si solar cell structure, such as passivated emitter and rear contact (PERC), heterojunction with intrinsic thin layer (HIT), and tunnel oxide passivated contact [1,2,3]. In industry, the structure of full-area aluminum-back-surface-field Si solar cell has been gradually replaced by a PERC structure solar cell [4,5,6,7]. For a PERC Si solar cell, the first step to make back contact at the rear side of a Si wafer with passivation layer stack is laser contact opening (LCO) [8,9,10], namely, using a laser to remove the passivation layer. Then, a full area Al layer is screen-printed. In this way, good quality contact can be achieved at the LCO area [11,12,13].

Local contact openings in a PERC solar cell can be made by printing etching paste [14], laser ablation, photolithography or mechanical scribing [15]. Laser ablation has been considered to be one of the best candidates for fabricating an industrial PERC cell. Laser processing eliminates physical contact, thus minimizing the probability of contamination. Laser processing speed can be extremely fast enabling high-throughput processing. This is unattainable by mechanical techniques such as mechanical scribing [16].

For both PERC and bifacial PERC solar cells, low rear-side contact resistivity between paste and Si-substrate is necessary for the fabrication of a high efficiency solar cell. Good contact between paste and substrate depends on many factors, such as Si substrate doping type, LCO size, metallization process and Al paste [17,18,19,20,21,22]. Picard E et al. studied the metallization condition and its effect on the voids formation at the LCO which affect solar cell V_oc_. A V_oc_ value of the PERC cells was reached at 665 mV. Better electrical performance parameters are limited by the front-side surface and the Si-bulk recombination [20]. Rauer M et al. demonstrated that contact penetration into the Si bulk can be reduced and the thickness of the Al-p^+^ region in the LCO area can be enhanced by intentionally adding some Si particles to the Al paste [23].

According to our results, we demonstrate that metallization on the rear side of a PERC solar cell can be efficiently achieved by the fire-through contact (FTC) method [24]. The passivation stack at the rear side of a PERC solar cell can be etched and/or reacted with Al paste and form a good back contact without a laser opening process, and the progress of making PERC solar cells would be much easier and cheaper [24,25,26,27]. The integration of the FTC approach into metal wrap through solar cells is presented in Thaidigsmann’s research, different FTC contact geometries and printing approaches are investigated [24]. Dominik Rudolph et al. developed an aluminum firing-through paste for the rear side metallization of p-type bifacial multicrystalline solar cells. The experimental results showed that the boron doping underneath the passivation stack is helpful to reach a good passivation quality and contact formation.

In this work, the firing-through paste can efficiently etch or react with the passivation stack and form a local back-surface field (BSF), and achieving a low specific contact resistivity ρ_c_ for 1.1 mΩ·cm^2^ with a V_oc_ for 647.3 mV. We also demonstrate that low-resistance rear side metallization can be achieved by first applying a glass paste layer on the rear side of a PERC solar cell to effectively react with the passivation layers, and Al paste can then be applied on the etched area. Point contact formed by firing through process can further reduce contact formation uniformity and improve PERC solar cell efficiency. In this study a p-type PERC solar cell with an area of 7.8 × 7.8 cm^2^ with a V_oc_ of 653.4 mV and an efficiency of 19.61% was fabricated.

## 2. Materials and Methods 

Semi-finished PERC solar cells with area of 78 × 78 mm^2^ were fabricated by Trina Solar Limited Company. The PERC solar cell has a structure of SiN_x_(80 nm)/n^+^-Si/p-Si/Al_2_O_3_(10 nm)/SiN_x_(80 nm). Cz–Si p-type wafers with a resistivity of ~1 Ω·m and a thickness of ~150 μm was employed for solar-cell fabrication. The SiNx and Al_2_O_3_ layers were prepared by plasma-enhanced chemical vapor deposition (PECVD). In this study, laser opening was not applied at the Si cell back side.

The firing-through pastes made in this study was composed of organic binder, glass frit, and Al powder or Al–Si eutectic metal powder. First, the paste components were mixed in a plastic jar, then the plastic jar was put into a high-speed centrifugal machine to mix the paste well. The mixed materials were dispersed on a roll mill to fully mix together the components of the paste. In this study 4 types of pastes were fabricated, and we named them paste 1, 2, 3, and 4. Paste 1 and 2 were made of the same glass frit using different conductive powder, Al-Si eutectic metal powder for paste 1 and Al powder for paste 2, respectively. Paste 2, 3 and 4 were made of the same Al powder component but with three different kinds of glass frits. The glass frits in paste 2, 3 and 4 were made of the same oxides using different weight ratio among the oxides. The oxides in the glass frit were PbO, SiO_2_, ZnO, Li_2_CO_3_, Cs_2_CO_3_, Sb_2_O_3_ and H_2_BO_3_.The different weight ratio among the oxides can get glass powder with different reaction ability. The main component of change was PbO and SiO_2_, and the best percentage of PbO and SiO_2_ in the glass frit was PbO: 50–60 wt.%, SiO_2_ 5–10 wt.%. The firing-through paste was screen-printed on the back side of Si PERC solar cells and then dried for 3 min at 200 °C in an oven in the atmosphere. The solar cell front Ag grid was then screen-printed, and finally the finished PERC solar cells were fired in a belt furnace with a peak temperature at ~750 °C.

The solar cell current-voltage (J-V) curves were measured under the standard AM1.5 illumination (1 kW/m2, 25 °C) on a solar J-V tester (IVT Solar Pte Ltd., Shanghai, China). The contact resistivity was measured by transfer-line measurements (TLM) on 10-mm-width stripes with equidistant metallization lines. The morphology of the pastes and the microstructures of the cells were all characterized by using a field-emission scanning electron microscope (FE-SEM, Sirion 200, HITACHI, Tokyo, Japan). The firing through reaction phenomenon at the paste/SiN_x_/Al_2_O_3_/Si interface was studied by using micro-Raman scattering. Raman spectra were recorded in a backscattering configuration using a Renishaw Micro-Raman spectrometer (Blue Scientific, Cambridge, UK). The laser lines 532 nm (solid state green laser) was used as the excitation sources. The laser light spot was fixed to a size of about 1 μm.

## 3. Results

Currently the chemical composition of an aluminum paste used for full-area back side metallization on a Si wafer cannot fire through the passivation stack, which is usually composed of an Al_2_O_3_ and a SiN_x_ layer. It is necessary to develop paste that can fire through the Al_2_O_3_/SiN_x_ bilayer stack without laser opening. Figure 1a shows the PERC solar cell back-side metallization fabrication process. For the fabrication of conventional bifacial Si PERC solar cell, the Al_2_O_3_/SiN_x_ passivation layer stack at the back side of a PERC solar cell is first removed by a laser-opening process, and then Al paste is screen-printed on the laser opening area. Then the solar cell module was heat treated at a relatively high temperature to achieve contact metallization and form a back surface field. In this study firing-through paste was designed to react with the Al_2_O_3_/SiN_x_ stack and then form a back surface field layer in the Si substrate, as shown in Figure 1b. In this case, the laser opening was not needed. This simplifies the cell fabrication process and would reduce the solar-cell fabrication cost. Figure 2a shows the pattern of firing-through paste screen-printing on the rear side. For this study, Figure 2b shows the photograph of a finished solar cell with back contact fabricated by using firing-through paste which was screen-printed on the cell rear side.

Firing-through process between Al paste and SiN_x_ layer on a Si substrate mainly depends on the reactions between the glass frit and the SiN_x_ layer at relatively high temperature during the fabrication process of a Si solar cell module. Figure 3a,b show the comparative surface morphologies of two cell Si surfaces after the paste was removed. The two wafers used were first screen printed with the Al paste developed in this study and the conventional PERC Al paste, respectively. The printed fingers had a width of 80 μm. Then, the two wafers were heat treated at a peak temperature of 750 °C. After the heat treatment, the two wafers were etched in diluted HCl to etch away the Al paste. From Figure 3b it can be seen that relatively heavy reactions have been occurred between the Si/Al_2_O_3_/SiN_x_ and the paste. The Si surface coated with the conventional PERC Al paste demonstrated almost no detectable etching effect, as shown in Figure 3a.

In this study, four series of aluminum pastes with different etching capability were developed. The Si wafer surfaces after firing-through reaction with three different types of paste are shown in Figure 4. It can be seen that the firing-through reaction between the paste and the Si/Al_2_O_3_/SiN_x_ depended strongly on the different components of the pastes. For the paste 1 and paste 2, the reacted surface area accounted for about 85% and 30% of the whole Al_2_O_3_/SiN_x_ stack surface, respectively. For the least reactive paste 4, only sporadic areas had been fired through, as shown in Figure 4e. Element mappings for the reacted areas, which demonstrated “white” color area in Figure 4a, are shown in Figure 5. It can be seen that in the reacted area, the element N has almost disappeared, as shown in Figure 5c, indicating that in this area the SiN_x_ was almost reacted with the paste. For the Al_2_O_3_ passivation layer, though with a lower intensity compared to that detected at the neighboring unreacted area, the Al EDS mapping still showed rather intensive signals at the reacted area. This means that a significant part of the Al_2_O_3_ passivation layer remained unreacted. This was confirmed by the following discussions.

The reaction between the firing-through paste and the Si/Al_2_O_3_/SiN_x_ structure was further studied by Raman scattering measurements, which were taken near the interface of paste/Si wafer. Raman spectroscopy is a sensitive and non-destructive method that can reveal information at atomic scale based on crystalline lattice vibrations. Figure 6 shows the optical microscopy images and the corresponding Raman scattering near the three paste/Si interfaces with different paste types, namely, paste1, paste 2 and paste 4. The Raman line “5” was taken in the Si wafer near the Si/paste interface. The detected areas in all the three samples were marked by the numbers “5” in the Si interior bulk wafers. If the detection location was far away from the Si/paste interface, the Raman scattering would only reflect the Raman scattering from the bulk Si lattice, namely, the typical phonon-scattering peak of the Si 1LO peak near 520 cm^–1^. The relatively broad Raman line “1”, which was detected in the paste far away from the Si/paste interface, showed a rather broad peak around the Si 520 cm^–1^. This peak originated in the Si diffused from the Si substrate during the firing-through reaction. The line “1” demonstrated a decreased intensity from paste 1, to paste 2 and paste 4. This observation was consistent with the microstructure observations shown in Figure 4. The Raman lines “2”, “3” and “4” were taken at different locations along the Si/paste interface. If the firing through reactions at the Si/paste interface were uniform, then the Raman scattering lines “2”, “3” and “4” would be the same, namely, they should be overlapped. For the most reactive paste 1, shown in Figure 6b, the three Raman lines “2”, “3” and “4” showed a relatively large difference both in Raman frequency and line width. The difference among the Raman lines “2”, “3” and “4” also indicates that at the Si/paste interface the interdiffusion/reactions were not uniform in a micro scale. In contrast, for the least reactive paste 4, shown in Figure 6f, the three Raman lines “2”, “3” and “4” almost overlapped with each other, indicating a rather uniform Si/paste interface. As shown in Figure 4c, the firing through reaction for the paste 4 was much less than that of the paste 1 as shown in Figure 4a,b.

The difference among the Raman lines “2”, “3” and “4” also indicates that at the Si/paste interface the interdiffusion/reactions were not uniform in a micro scale. This was confirmed by the high-resolution SEM observations shown in Figure 7. Figure 7 shows the Si/paste 1 interface after firing-through reaction in different micro-scale resolution. It can be seen that about 85% of the Si/paste interface had been reacted after the firing-through process. In the heavily reacted region, it was found that part of the Al_2_O_3_/SiN_x_ did not react with the Al paste. Figure 7b shows that in the heavily reacted region some part of the Al_2_O_3_/SiN_x_ did not react with the Al paste, rather it was broken due to the reaction of Al with the Si underneath the Al_2_O_3_/SiN_x_ layer. The uneven ablation area would increase recombination of carriers and reduce the open circuit voltage.

A back-surface field layer formed by alloying of Al with Si was detected by electrochemical capacitance-voltage (ECV) measurements. Localized component detections around a spike, which is about 5 μm deep into the Si, were carried out to quantitatively characterize the components of Si and Al. The EDX data showed that near the spike/Si interface area, significant alloying between Al and Si had occurred. This observation was confirmed by the ECV measurement [21,22] shown in Figure 8. 

Figure 8 shows two Al doping profiles obtained by ECV measurement of a normal full-area aluminum back field cell and a PERC cell fabricated in this study by using Al paste 1. The full-area aluminum back field cell had no passivation stack layer at the cell back side. Therefore, the reaction between Al and Si was fully reacted, resulting in a relatively Al-high doping depth profile of around 5 μm into the Si. For the PERC cell with firing through paste 1, it can be seen that the Al doping depth of paste 1 was rather shallow, and the thickness of the BSF was only around 1 µm. For both the cells, the doping concentrations at the depth range from the surface to ~1 μm into the Si substrates were at the same level, namely, ~1.5 × 1019 cm^–3^. Such a doping concentration can satisfy the requirement of low contact formation on a Si substrate [20]. From Figure 8 it can be seen that the Al depth and doping uniformity obtained by using the firing-through technique in this study are needed to be further improved compared to the currently dominant full-area Al-BSF paste solar cells. 

The J-V curves of the four p-type Si solar cells fabricated with the four different types of Al pastes are shown in Figure 9. The specific contact resistivity ρ_c_ is 1.1, 55.4, 124.3, and 234.1 mΩ·cm^2^ for the four Si cells with pastes 1 to 4, respectively. For the most reactive paste 1, it can be seen that a low resistivity ρ_c_ in the order of 1.0 mΩ·cm^2^ was successfully achieved. Figure 9b shows the dependence of the fill factor FF and the open-circuit voltage V_oc_ on the contact resistivity. With a low specific contact resistivity, the solar cell fill factor was increased. However, as discussed above and shown in Figure 5, for the solar cell with the most reactive paste 1, the open-circuit voltage was 6 mV lower than that of the cell with the paste 4. This was induced by the increased carrier recombination at the Si/paste interface. For the Si cell fabricated with paste 4, the Si/paste interface was rather uniform, but the firing-through reaction between the passivation stack and the paste was far less than that of the cell with paste 1. This led to a much high specific contact resistivity ρ_c_ of 234.1 mΩ·cm^2^ and low fill factor of 70.1%.

It can be seen from this study that the paste 1 is the most reactive paste to fire through the passivation stack. In the Al paste, the active component to etch the passivation stack was the glass frit. In order to enhance the fire-through etching process, we have made glass pastes, in which Al was not added. In this case, the glass frit would be more intensely and physically contacted with the passivation stack, while for Al paste, a significant part of the surface of a passivation stack would be occupied by the Al particles in the paste. Al particles are not the reactive component for the fire-through process. Therefore, glass frit components would be more critical for Al paste to fire through the passivation stack. In order to demonstrate the etching effect of glass frit on solar cells, two types of glass pastes, in which no Al was added, were prepared and screen printed on two PERC Si solar cells. The glass frit components in the two glass pastes were the same as that of the glass frit in the paste 1 which makes sure an efficient etching reaction between the glass frit and the passivation stack, as discussed above. The difference between the two glass pastes made in this study was only that the solid glass frit was 10% and 30% of the total glass paste, respectively. Figure 10a,b show the surface morphologies of two fingers made with the two different glass pastes after firing in a furnace with a peak temperature around 500 °C. As shown in Figure 10b, the clear edges between the glass paste and the Si substrate demonstrated that a significant reaction had occurred between the glass frit and the passivation stack. After the heat treatment, a full area Al paste layer was screen printed on the two solar cells and the cells were fired in a furnace. The peak heat treatment temperature was 750 °C. In order to verify directly whether the firing through reaction has occurred, the two Si cells were then etched in HCl to remove the Al fingers. Figure 10c,d show the finger morphology after the finger removal. It can be seen that the finger area was uniformly fired through for the glass paste with 30% solid glass as shown in Figure 10d. For the finger fired with 10% glass frit, the finger was not uniformly fired through, as shown in Figure 10c. However, the etching reaction was still significant. This was confirmed by the J-V curve measurements, shown in Figure 11a,b. The solar cell short circuit currents were 39.71 and 39.60 mA/cm^2^, the V_oc_ were 0.6479 and 0.6401 V, the fill factors were 74.6% and 74.0%, and the efficiency were 19.19% and 18.76%, for the two solar cells fabricated with 30% and 10% solid glass frit, respectively. It can be seen that the V_oc_ of the cell using the 10% glass frit was 7.8 mV lower than that used the 30% glass frit. This was induced by a relatively high density of spikes observed in the Si for the 10% glass frit cell, as shown in Figure 10e. The spikes ran deep into the Si, therefore increasing carrier recombination around the spikes leading to the loss of its V_oc_. For the 30% glass frit cell, the reaction between the Al paste and the passivation stack was more uniform compared to the 10% glass frit cell, as shown in Figure 10f. This observation was also confirmed by the dark J-V measurements shown in Figure 11b. The 10% glass frit cell showed an increased leakage current at the reverse applied voltage.

In order to reduce carrier recombination at the rear surface of PERC solar cells, point contacts were designed and fabricated with the same paste-etching processes as discussed in Figure 10. The solar cell contact structure and the light J-V curve are shown in Figure 12. The solar cell demonstrated a high open-circuit voltage of 0.6534 V, the fill factor was 73.1%, and the efficiency was 19.61%. The dark J-V curves of the cells with screen printing firing-through line paste 1 and point contact are shown in Figure 12b. It can be seen that the Si cell with point contact had lower leakage current, which was ascribed to the reduced rear surface recombination compared to the firing-through paste with line shape. Figure 12e shows the cross-sectional SEM image of one point contact. It was found that 80% of all the point contacts showed such an inverted pyramid shape. Small contact area and uniform etching for the point contact solar cell led to relatively high open-circuit voltage of 0.6534 V. This V_oc_ value is one of the highest values for all the firing through solar cells fabricated in this study. The fill factor was relatively low, 73.1%, which was induced by the relatively high series resistance due to the small contact area shows in the Figure 12c.

## 4. Conclusions

In this study, firing-through pastes with different etching capability have been developed and have been applied to achieve metallization contact on the rear side of p-type monocrystalline silicon PERC solar cells. The rear-side passivation layer Al_2_O_3_ and the SiN_x_ layer can be effectively etched by the firing-through paste. Ohmic contact with a contact resistivity as low as 1 mΩ·cm^2^ was obtained for the most reactive paste. However, when the metallization contact resistance was low, the open-circuit voltage was also lower than that of the cells with higher metallization contact resistance. A non-uniform etching reaction between the paste and the Al_2_O_3_/SiN_x_ would lead to formation of high-density recombinations at the Si/paste interface. Small-area patterned dot back contacts can improve the metallization uniformity during the fire-through process. Even though the majority of the dot contacts showed inverted pyramid contact geometry, carrier recombination was reduced due to the small-contact area at the Si/paste interface. A p-type solar cell with an area of 7.8 × 7.8 cm^2^ with a V_oc_ of 0.6534 V and an efficiency of 19.61% was fabricated.

## Figures and Tables

**Figure 1 materials-12-03388-f001:**
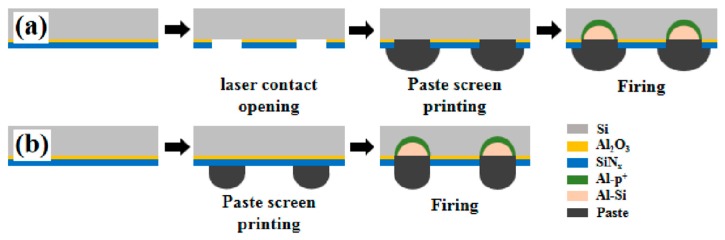
(**a**) Progress flow for passivated emitter and rear contact (PERC) cell fabrication. In this case contact metallization was assisted by laser opening process; (**b**) progress flow for PERC cell fabrication by using firing-through paste. In this case laser opening process is not necessary.

**Figure 2 materials-12-03388-f002:**
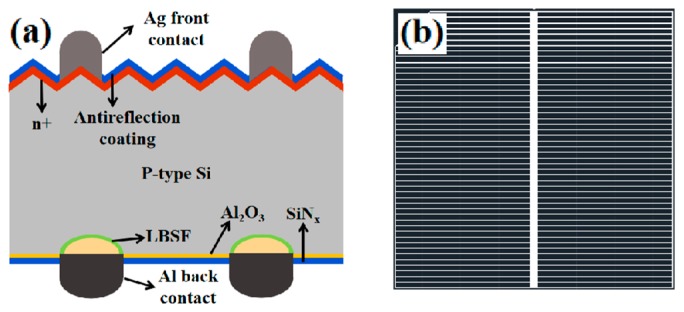
(**a**) Schematic structure of the solar cell fabricated in this study; (**b**) the pattern of firing-through paste screen-printing on the rear side.

**Figure 3 materials-12-03388-f003:**
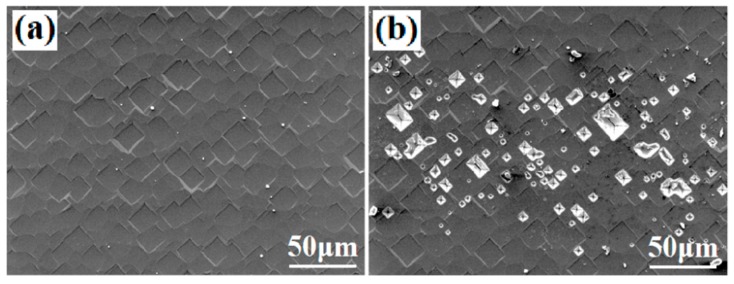
The surface morphologies two cells after the contact pastes were etched away. (**a**) Cell with conventional PERC Al paste; (**b**) cell with firing-through Al paste.

**Figure 4 materials-12-03388-f004:**
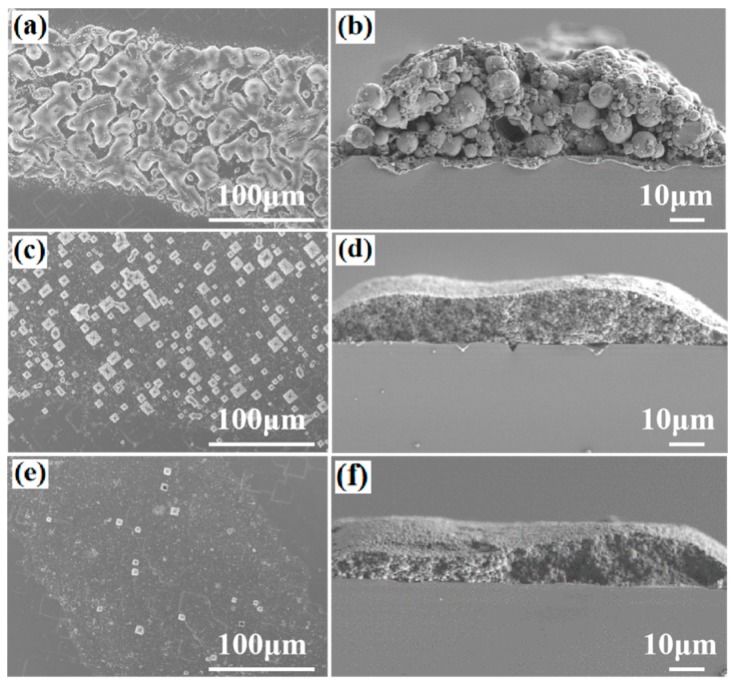
(**a**,**c**,**e**) Surface morphologies of cells with paste 1, 2, 4 after the contact pastes were etched away; (**b**,**d**,**f**) cross section of contact grid lines fabricated with paste 1, 2, 4.

**Figure 5 materials-12-03388-f005:**
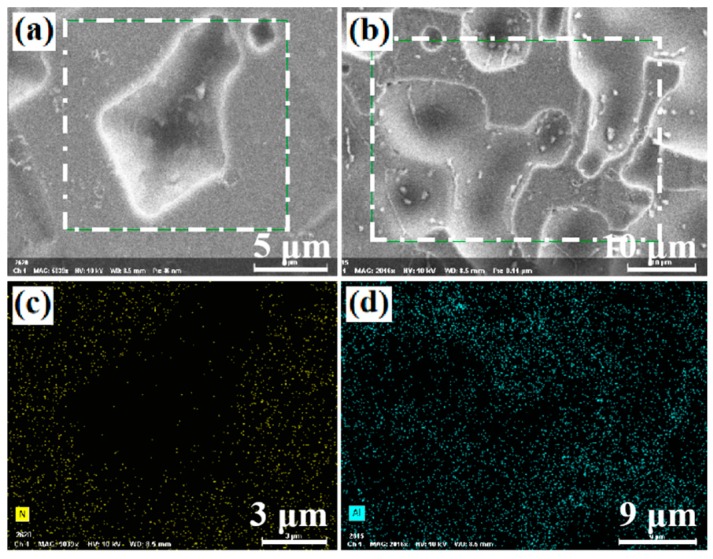
(**a**,**b**) scanning electron microscope (SEM) surface images of the reacted areas after the Al paste 1 had been etched; (**c**,**d**) element mapping for nitrogen and Al at the reacted areas shown in (**a**,**b**), respectively.

**Figure 6 materials-12-03388-f006:**
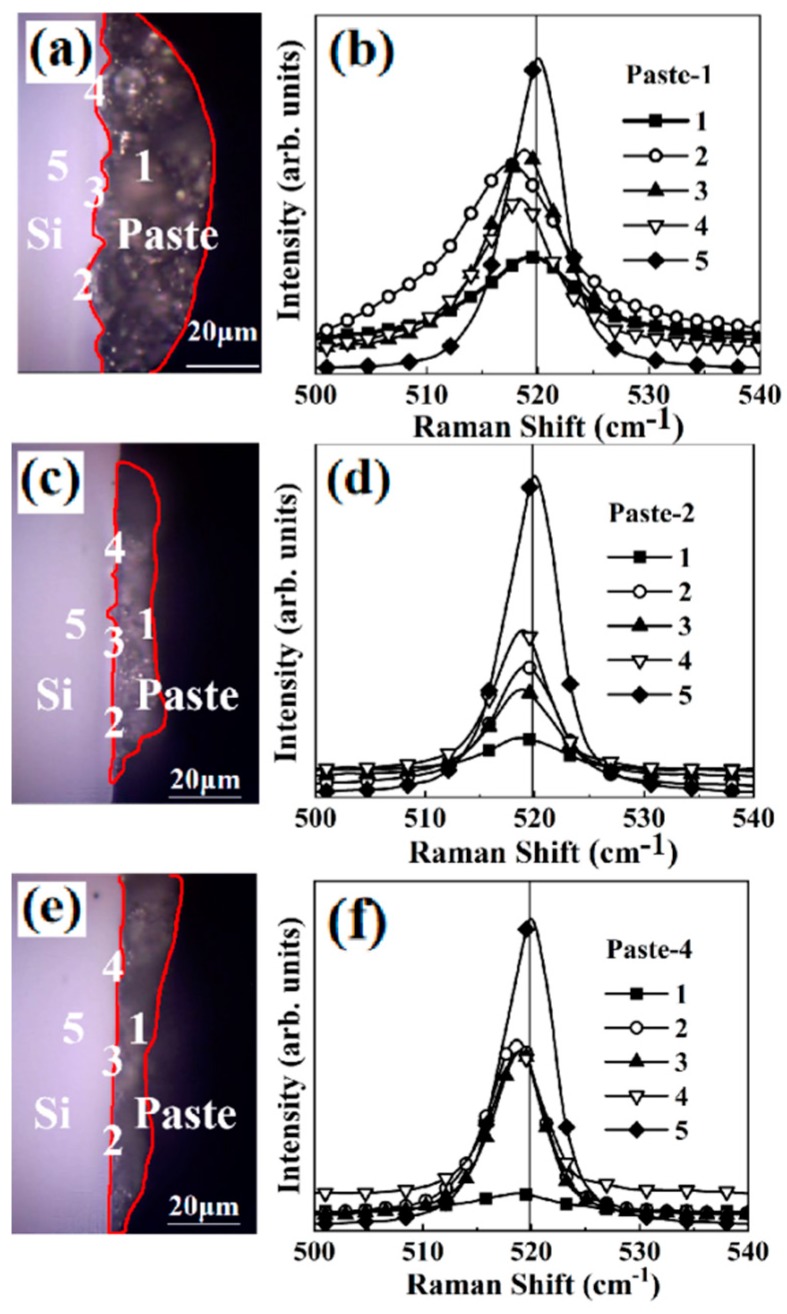
(**a**,**c**,**e**) Optical microscope images of Si/paste interfaces with paste 1, 2 and 4; (**b**,**d**,**f**) Raman spectra of the Si 1LO detected at different positions near the Si/paste interface. The detected positions are marked as numeric numbers from 1 to 5 in the optical microscope images.

**Figure 7 materials-12-03388-f007:**
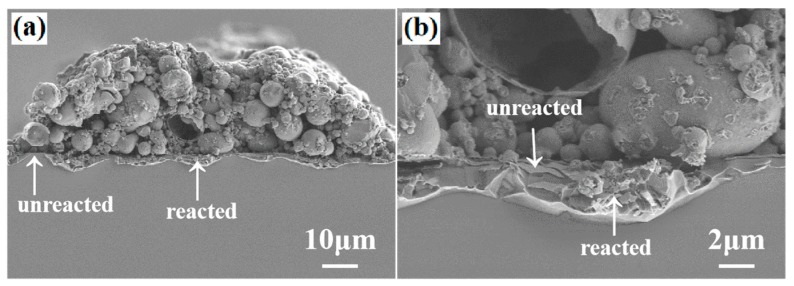
(**a**) SEM image at the Si/paste 1 interface after firing through reaction; (**b**) detailed SEM image showing non-uniform reactions at the Si/paste 1 interface.

**Figure 8 materials-12-03388-f008:**
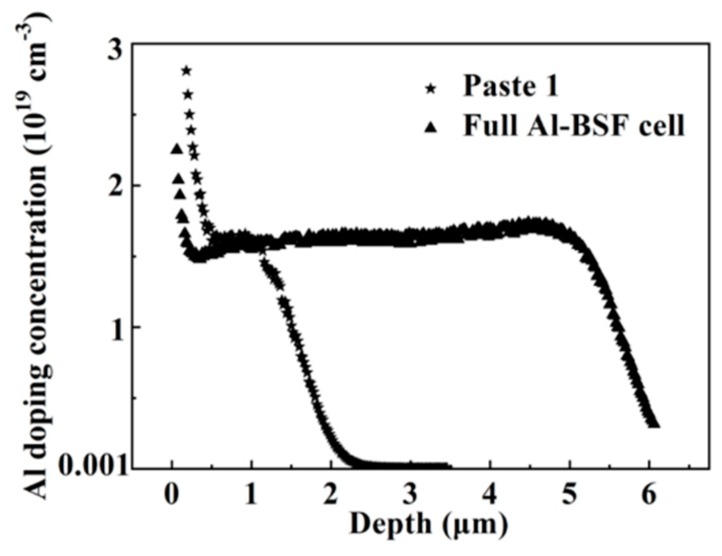
Comparative electrochemical capacitance-voltage (ECV) measurements of the doping profiles obtained in the Si substrates using the firing-through paste 1 and normal full-area Al-back-surface field (BSF) paste, respectively.

**Figure 9 materials-12-03388-f009:**
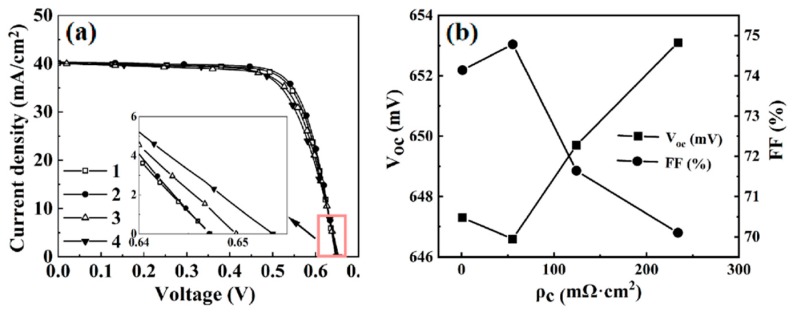
(**a**) J-V curves of four p-type Si solar cells fabricated with the four different types of Al pastes; (**b**) dependence of the fill factor and the open-circuit voltage on the contact resistivity.

**Figure 10 materials-12-03388-f010:**
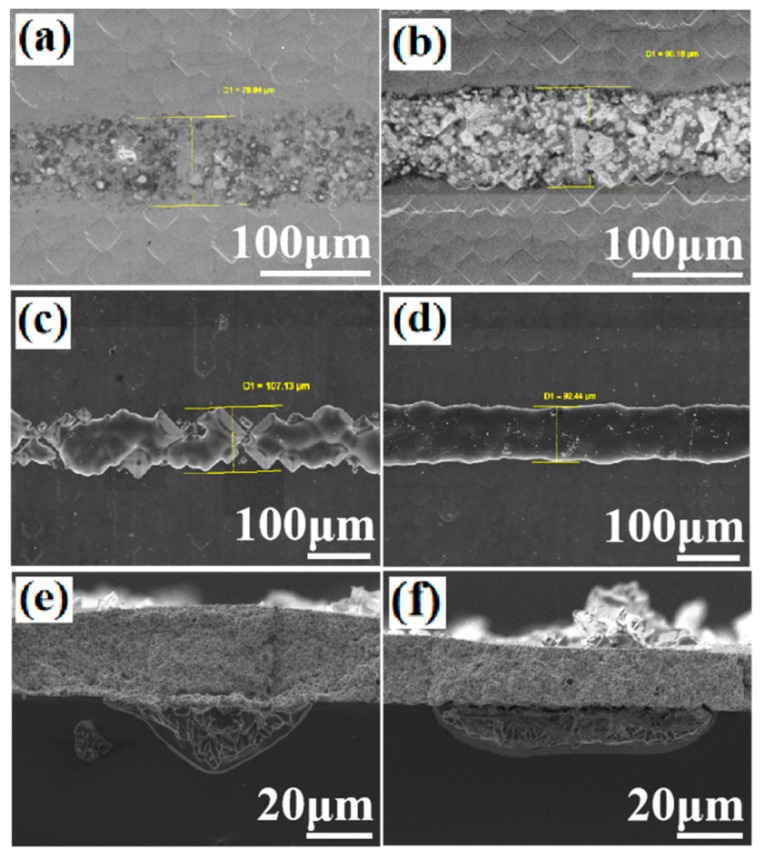
(**a**,**b**) SEM surface images obtained using the 10% and 30% solid glass frit in the glass pastes; (**c**,**d**) SEM images of the etching area after the Al paste had been removed; (**e**,**f**) cross-sectional SEM images of the contact area.

**Figure 11 materials-12-03388-f011:**
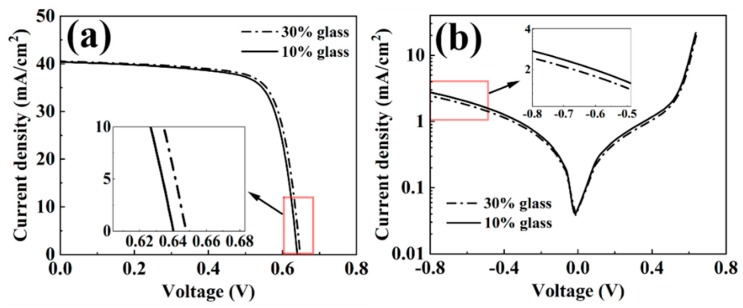
(**a**,**b**) Light and dark J-V curves of the cells with 10% and 30% solid glass frit in the glass pastes.

**Figure 12 materials-12-03388-f012:**
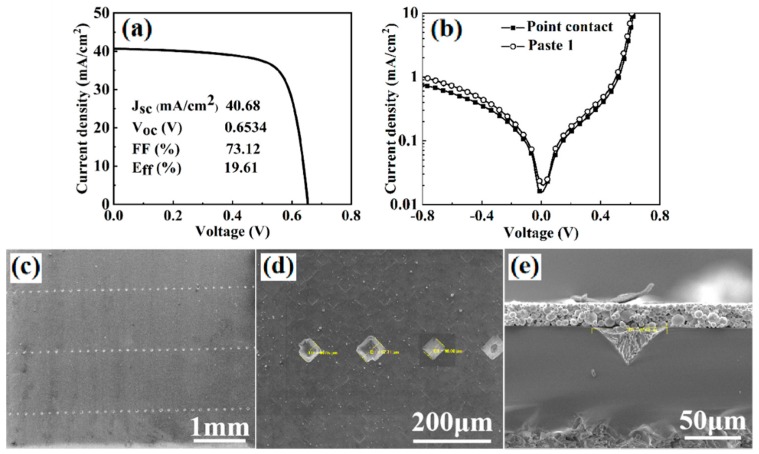
(**a**) Light J-V curve of a solar cell with point contact; (**b**) dark J-V curves of the cells obtained with point contact by using paste 1; (**c**,**d**) SEM surface images of point contact cell and (**e**) cross-sectional SEM image obtained on one-dot contact.

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
