# Peer review of "High-Efficiency p-Type Si Solar Cell Fabricated by Using Firing-Through Aluminum Paste on the Cell Back Side"

_materials, 2019, doi:10.3390/ma12203388_

Round 1

Author Response

Dear Editor,

We have studied the valuable comments from you, the assistant editor and reviews carefully, and tried our best to revise the manuscript. The point to point responds to the reviewer’s comments are listed as following:

Responds to the reviewer’s comments:

Reviewer 1

Comment 1: Since the paragraphs in line#52 and 62 have started in the same way, I recommend to revise or paraphrase the presentation of paragraph in line #52.

Response: Thank you for your valuable advice. “In this work” in line #52 has been revised into “According to our results”.

Comment 2: It has mentioned that the glass frits in paste 2, 3 and 4 were made of the same oxides using different weight ratios among the oxides. The oxides in the glass frit were PbO, SiO2, ZnO, Li2CO3, Cs2CO3, Sb2O3, and H2BO3. Why do you use a different weight ratio? Which ratio is best?

Response: Thanks for your question. The different weight ratio among the oxides can get glass powder with different reaction ability. The main component of change is PbO and SiO2, PbO and SiO2 in the glass frit PbO: 50~60 wt%, SiO2 5~10 wt%. These information have been added in the manuscript.

Comment 3: Is there any elemental diffusion from the paste which causes device degradation?

Response: Thanks for your question. The element of Fe in the paste will cause device degradation, but the content of Fe is very low, and the influence can be ignored.

Comment 4: It is difficult to read inset in Fig. 10. Please revise it.

Response: Thanks for your question. We have revised the Fig. 10 to two figures for clear presentation.

We have tried our best to improve the manuscript. We appreciate for Editors/Reviewers’ warm work earnestly, and hope that the correction will meet with approval.

Once again, thank you very much for your comments and suggestions.

Reviewer 2 Report

Line 72. Lack of  wafer resistivity.

Line 96. Mistake "miro-Raman"

Line 99. Lack of Pitch between laser spots

Line 119. "Fig. 2(b) shows the photograph of a finished solar cell". The picture shown does not correspond with a finished solar cell.

Line 129."From Fig. 1(b) it can be seen that relatively heavy...". The reference to the figure is wrong.

Line 135. "In this study four series of aluminum paste..." and in line 136 you talked about three different types of paste (paste 1,2,4) showing the surface morphologies in Fig.4.Please, clarify or include the experiments for paste 3.

Line 196. results using EDX  and ECV are metioned, but only results of ECV are shown.

Author Response

Dear Editor,

We have studied the valuable comments from you, the assistant editor and reviews carefully, and tried our best to revise the manuscript. The point to point responds to the reviewer’s comments are listed as following:

Responds to the reviewer’s comments:

Reviewer 2

Comment 1: Line 72. Lack of wafer resistivity.

Response: Thank you for your valuable advice. The information of wafer resistivity has been added in line 73.

Comment 2: Line 96. Mistake "miro-Raman"

Response: We are sorry for our mistake. “miro-Raman” has been revised into “micro-Raman”.

Comment 3: Line 99. Lack of Pitch between laser spots

Response: Thank you very much. We do not have a fixed spacing between signal acquisition points on the sample, we randomly selected different positions of the sample for testing while using micro-Raman scattering.

Comment 4: Line 119. "Fig. 2(b) shows the photograph of a finished solar cell". The picture shown does not correspond with a finished solar cell.

Response: We are sorry for our mistake. “Fig. 2(b) shows the photograph of a finished solar cell” in line 122 has been revised into “Fig. 2(a) shows the pattern of firing-through paste screen-printing on the rear side”.

Comment 5: Line 129."From Fig. 1(b) it can be seen that relatively heavy...". The reference to the figure is wrong.

Response: We are sorry for our mistake. “From Fig. 1(b)” in line 132 has been revised into “From Fig. 3(b)”.

Comment 6: Line 135. "In this study four series of aluminum paste..." and in line 136 you talked about three different types of paste (paste 1,2,4) showing the surface morphologies in Fig.4.Please, clarify or include the experiments for paste 3.

Response: Thanks for your question. The performance of the glass frit used in paste 3 is similar to the glass frit used in paste 4, therefore the electrical performance of solar cell which screen-printing the paste 3 or the paste 4 is close, in order to save space, we did not present the result of paste 3.

Comment 7: Line 196. results using EDX and ECV are metioned, but only results of ECV are shown.

Response: Thanks for your question. The data of EDX could not accurately express the proportion of elements of BSF. And the profile of ECV can represent the doping concentration of BSF. So, we only show the results of ECV. The information about EDX has been deleted in line 199.

We have tried our best to improve the manuscript. We appreciate for Editors/Reviewers’ warm work earnestly, and hope that the correction will meet with approval.

Once again, thank you very much for your comments and suggestions.